# Update on Non-Pharmacological Interventions for Treatment of Post-Traumatic Headache

**DOI:** 10.3390/brainsci12101357

**Published:** 2022-10-06

**Authors:** Matthew J. Lee, Yi Zhou, Brian D. Greenwald

**Affiliations:** 1Hackensack Meridian School of Medicine, Nutley, NJ 07110, USA; 2JFK Johnson Rehabilitation Institute, Edison, NJ 08820, USA

**Keywords:** post-traumatic headache, PTH, headache, non-pharmacological, concussion, traumatic brain injury, mTBI

## Abstract

Post-traumatic headache (PTH) is the most common sequelae of traumatic brain injury (TBI). Its phenotypic variability, absence of formal evidence-based guidelines for treatment and underdiagnosis have made its management a challenge for clinicians. As a result, treatment of PTH has been mostly empiric. Although analgesics are the most popular drug of choice for PTH, they can present with several adverse effects and fail to address other psychosocial comorbidities associated with TBI. Non-pharmacological interventions thereby offer an intriguing alternative that can provide patients with PTH sustainable and effective care. This review article aims to: (1) provide an update on and describe different non-pharmacological interventions present in the recent literature; (2) provide clinical guidance to providers struggling with the management of patients with PTH; (3) emphasize the need for more high-quality trials examining the effectiveness of non-pharmacological treatments in patients with PTH. This review discusses 21 unique non-pharmacological treatments used for the management of PTH. Current knowledge of non-pharmacological interventions for the treatment of PTH is based on smaller scale studies, highlighting the need for larger randomized controlled trials to help establish formal evidence-based guidelines.

## 1. Introduction

Traumatic brain injury (TBI) is a leading cause of disability in the United States today. According to the Center for Disease Control and Prevention (CDC) [1], there were an estimated 223,135 hospitalizations attributed to TBI in 2019, which may be an underestimation given that many cases often go unreported [2,3]. In the United States, TBI is most commonly caused by falls, motor vehicle crashes and recreational events (e.g., sports) [2]. After injury, individuals may experience acute and chronic complications with post-traumatic headache (PTH) being one of the most common [4]. It is estimated that 14–58% of individuals will experience PTH within the year following a TBI [5,6].

According to the International Classification of Headache Disorders, 3rd Edition (ICHD-3), PTH is defined as a secondary headache that develops within 7 days of trauma to the head or within 7 days of regaining consciousness. Post-traumatic headache can be subcategorized as “acute” if headache symptoms last less than 3 months and “persistent” if they last longer than 3 months. Although categorized as a single disorder, PTH does not have a uniform presentation and can manifest with variable headache phenotypes. Individuals with PTH commonly report migraine and tension-type headache phenotypes; however, those resembling cervicogenic headache, cluster headache, hemicrania and other primary headache disorders have been noted as well [7,8,9]. PTH will often have overlapping features of several headache phenotypes or be deemed as non-classifiable, demonstrating the complexity of the disorder and the need for a comprehensive patient history. While headache may be the primary symptom of PTH, it may also present with a combination of cognitive and emotional deficits. Post-traumatic headache has been associated with post-concussive syndrome symptoms such as dizziness, sleep disturbances and mood disorders [8,10].

Several potential risk factors for PTH have been investigated as well. Female gender, adolescent age, prior headache history and pertinent family headache history have been associated with an increased risk of developing PTH following TBI [5,10,11]. Studies have also found there to be an inverse relationship between TBI severity and PTH development. A higher rate of PTH cases were found following mild TBI (mTBI) events compared with more severe injuries [3,5,9].

Athletes are another group that are at increased risk for PTH and other TBI complications. While dependent on the sport, athletes are more likely to experience direct impacts to the head from collisions, falls or accelerating objects [12]. They are also at risk of chronic neurological complications due to premature return to play. Athletes may disregard signs of head trauma or demonstrate reluctance in reporting symptoms in order to continue playing [3]. As headache is the most common sequelae of TBI, it is vital that symptoms are recognized in a timely manner to avoid morbid complications from repeated trauma. If not, repetitive injuries to the head may lead to prolonged PTH pain, depression and potential progressive neurological deterioration [2].

Although pro-inflammatory mechanisms have been suggested through animal trials, the pathophysiology of PTH remains unclear, making its management a complex challenge [12]. Surveys conducted among clinicians have demonstrated varied opinions regarding the treatment of PTH and there are currently no formal evidence-based guidelines to direct providers [13,14,15]. Analgesics are the most popular pharmacological intervention used in the management of headache symptoms, but their use may be limited when accounting for adverse effects and medical comorbidities. Given their properties, non-steroidal anti-inflammatory agents (NSAIDs) may reduce the potential inflammation and pain associated with PTH. However, persistent use can potentially lead to medication overuse headache (MOH), acute renal failure or gastrointestinal complications such as acute gastritis and gastric ulcers [16]. Acetaminophen avoids some of these adverse effects, but its potential hepatotoxicity and lack of anti-inflammatory action may be ineffective in PTH management. Aside from analgesics, other drugs used to treat PTH include antiepileptics and tricyclic antidepressants, which also present several drawbacks [17].

In addition to potential adverse effects, pharmacological interventions may also fail to address the comprehensive needs of patients with PTH [18]. For example, while NSAIDs can alleviate headache symptoms, they do not address concomitant functional and mood disturbances that can occur following TBI. This can lead to a delay in recovery and further health complications that can hinder quality of life. Non-pharmacological treatments thereby offer an effective alternative that limits any debilitating side effects and may also address vital psychosocial impacts of PTH [19,20]. There are a variety of these non-pharmacological interventions that may meet the particular and holistic needs of a patient with PTH (Table 1). The aim of this review is to provide an update on non-pharmacological interventions that have been used to treat PTH, guidance to clinicians struggling to treat patients with PTH and guidance on future research and trials.

## 2. Methods

The search strategy for this review consisted of an online search of PUBMED (Figure 1). The database was utilized to find references published from 2010 to 2022 and used key terms: “posttraumatic headache”, “PTH”, “headache after TBI”, “post concussive headache”, “treatment”, “management” and “therapy”. Boolean operators “AND” and “OR” were used to target and narrow the search results. Some references were found by using reference lists from other included articles. Our inclusion criteria were studies published from 2010 to 2022 that discussed PTH and its non-pharmacological treatments. Case studies and references primarily focusing on pharmacological interventions were excluded from this review.

## 3. Cognitive and Behavioral Modification

Twamley et al. [21] conducted a pilot randomized control trial (RCT) assessing the use of Cognitive Symptom Management and Rehabilitation Therapy (CogSMART) in veterans with TBI. One goal of the intervention was to alleviate post-concussive symptoms through psychoeducation, stress reduction, sleep hygiene and headache management. The study also targeted other domains including prospective memory, attention, learning and executive function through other cognitive modules (problem solving, journaling). The participants were followed over the course of 12 weeks and results showed a significant reduction in post-concussive symptoms including headache, dizziness and nausea. Participants also demonstrated a significant improvement in prospective memory performance at post-treatment.

Compared with cognitive training, behavioral therapy focuses on manipulating both cognitive and physical behaviors to promote healthy changes that can improve symptoms [22]. Progressive muscle relaxation (PMR) is an example of a behavioral intervention that has been studied in the treatment of headache. It is a technique that aims to reduce tension by tensing and relaxing one’s muscles in an alternative manner. Currently, there is limited evidence on the impact of this therapy on headache symptoms following TBI. Usmani et al. [23] conducted a longitudinal single-arm study examining the use of PMR in individuals with persistent PTH. Participants used a smartphone application to access the intervention and took part in 20 min of the exercise daily with recording of symptoms. Results showed that participants who used PMR more than 4 times per week demonstrated decreased incidence of headache. However, the authors did not determine statistical significance.

Cognitive behavioral therapy (CBT) is a specific type of behavioral treatment that integrates cognitive reframing of mental and physical behaviors. The strategy has become popular in the management of several psychosocial conditions and has been hypothesized as a potential treatment for PTH [24]. Although CBT has been investigated in patients with headache following TBI, the results have been mixed [8,25].

One quasi-experimental study [26] examined the effectiveness of a cognitive behavioral exercise approach (CBEA) in patients who experienced whiplash. Patients in the interventional group were provided psychoeducation about whiplash and its symptoms and prescribed neck exercises that were performed under the supervision of a physiotherapist. The program consisted of seven neck movement exercises, four neck strengthening exercises and three shoulder movement exercises. Patients in the CBEA group demonstrated a reduction in both headache and neck pain compared with the control group.

While there has been a limited number of RCTs examining the use of CBT in headache following TBI, Kjeldgaard et al. [27] investigated its use in adult patients with chronic PTH. The intervention consisted of a combination of cognitive education and relaxation techniques in the form of autogenic training and PMR. Although patients were followed for 26 weeks, there were no significant reductions in headache scores post-treatment.

McGeary et al. [28] implemented a triple-armed RCT that compared CBT for headache (CBTH), cognitive processing therapy (a PTSD-specific type of CBT) and the usual treatment in the management of military veterans with comorbid PTH and PTSD. After 8 weeks, the group receiving CBTH demonstrated a significant decrease in 6-item headache impact test (HIT-6) scores compared with those receiving the usual treatment, indicating an improvement in headache-related disability and quality of life. While HIT-6 scores were reduced in the group receiving CPT compared with the group receiving the usual treatment, the difference was not statistically significant.

Another RCT [29] was conducted specifically focusing on the impact of CBT in adolescents experiencing symptoms following concussion. A collaborative care intervention was implemented in which participants took part in CBT but also received holistic care from physicians across disciplines and subspecialists. The CBT consisted of pacing activities, coping skills, relaxation techniques and sleep hygiene. Patients took part in the study for 6 months and the intervention group showed significant improvement in post-concussive symptoms including headache.

## 4. Acupuncture

Acupuncture is a non-pharmacological technique that has been demonstrated to benefit patients experiencing migraines [30]. Jonas et al. [31] conducted a triple-armed, parallel, randomized controlled study exploring the impacts of both auricular acupuncture (AA) and traditional Chinese acupuncture (TCA) in patients with PTH compared with the usual care. The latter uses longer sessions and typically places a focus on energy balancing and the patient’s personality and weaknesses when contextualizing a patient’s care. In contrast, AA involves shorter sessions and utilizes a more systematic and sequential methodology of inserting needles and is guided by the patient’s response. Using a standardized treatment protocol, the authors recorded HIT-6 scores at 6 weeks. Post-treatment, there was a significant reduction in headache scores in groups receiving AA compared with those receiving the usual care. There was also a significant improvement in HIT-6 scores in patients receiving acupuncture (composite AA and TCA data) compared with the usual care. Both AA and TCA groups also showed significant reductions in current and best global pain (NRS) when compared with the usual care. There were similar results when the composite score for acupuncture was compared with the usual care [31]. While the study supports the potential use of acupuncture in the treatment of PTH, more high-quality trials are needed.

## 5. Lifestyle Modifications and Physical Therapy

Patient education on lifestyle modifications following TBI can potentially serve as an intervention to alleviate PTH symptoms and maintain headache health longitudinally [21,32]. Particular lifestyle changes that could benefit headache following TBI were identified in a study conducted by Baker et al. [33] The authors examined the use of lifestyle changes in combination with CBT and biofeedback in military personnel experiencing PTH over the course of 2 years. These modifications include an increased emphasis on oral rehydration (electrolyte drinks), improvements in sleep hygiene and stress reduction. At post-treatment, there was an observed 36% reduction in headache frequency and a 56% reduction in severity. Sixty percent of participants also reported an improvement in quality of life when compared to pre-intervention. While evidence is limited, this suggests that rehydration, sleep hygiene and stress reduction may be lifestyle modifications that can potentially improve PTH outcomes [32].

Exercise and physical therapy are other lifestyle modifications that have been suggested for patients experiencing PTH [9,11,34]. It is believed that physical activity can alleviate inflammation in the central nervous system and reduce oxidative stress that are related to headache. Similarly, physical therapy is thought to reduce migraine symptoms by maintaining the range of motion of neck muscles and removing cervicogenic triggers [9,35].

Wilson et al. [36] conducted a retrospective study investigating the impact of early physical activity on outcomes in pediatric patients following concussion. Patients who participated in some form of physical activity after injury but prior to evaluation were compared with those who did not. Upon review, the authors found that the presence of current headache was significantly less in those who participated in early physical activity. Results also demonstrated that patients who did not engage in early physical activity had longer symptom duration and increased odds of post-injury headache.

Another retrospective cohort study was performed by Grabowski et al. [37] examining the use of multimodal physical therapy in athletes with post-concussion syndrome. The authors hypothesized that a multimodal plan integrating sport-specific training with vestibular, oculomotor and cervical rehabilitation would improve PTH symptoms. At post-treatment, there was a reduction in the headache severity score reported on the post-concussion symptom scale (PCSS). The authors advocate that supervised physical therapy focusing on physiologic impairments may be beneficial.

A physical therapy regimen focused on cervicovestibular rehabilitation may also improve headache outcomes in patients following TBI. Following a traumatic event such as a concussion, PTH may stem from a trigger such as trauma to the cervical spine, nerve roots or caudal brainstem [38,39]. Given that migraine, dizziness and vertigo are common symptoms associated with PTH, the vestibular system could be a potential target for therapeutic interventions [40]. An RCT conducted by Schneider et al. [39] examined the use of a cervicovestibular physical therapy in sport-related concussions during an 8-week period. At the end of the study, a significantly greater number of patients in the treatment group were medically cleared to return to sport and reported no current headache. While the authors found the combined cervicovestibular approach promising, more research is needed to determine its impact on PTH.

## 6. Nutraceuticals

Nutraceuticals are food substances that can provide medical and health benefits. They are non-pharmacological alternatives that have been used for the management of migraines and other neurological conditions. Monsour et al. [41] suggest that their use may be beneficial in treating patients with PTH as described through their SHARED Model of Care. In this model, the authors propose lifestyle changes that promote headache health in youths, including taking nutraceuticals such as vitamin D, riboflavin (vitamin B2), coenzyme Q and melatonin. While such an intervention seems promising, there is limited clinical evidence supporting their use in headaches following TBI. Currently, there are no controlled trials investigating the effects of vitamin D, riboflavin or coenzyme Q in human subjects specifically with PTH; however, potential benefits and mechanisms have been described and examined in other studies.

Vitamin D, a compound found in several food items and dietary supplements, has been proposed for potential use in TBI patients given its anti-inflammatory properties. In one animal study, a high vitamin D diet in rats resulted in reduced inflammatory and TBI biomarkers, suggesting possible beneficial effects in the recovery of patients with PTH [42]. Riboflavin (vitamin B2) and coenzyme Q10 (CoQ10) have also been thought to play a role in migraine pathophysiology, playing a role in oxidative metabolism [43]. Animal studies examining riboflavin have shown reductions in cerebral edema and cortex injury [44]. Similarly, the administration of CoQ10 in animals demonstrated decreases in TBI markers, neuronal loss and brain mitochondrial damage, indicating its potential use in patients with PTH [45]. Further research is needed to determine whether these nutraceuticals have significant effects in PTH.

Melatonin is a hormone released by the pineal gland that can be exogenously obtained through diet and supplements. Given its safety and lack of major side effects, melatonin may serve as a potential non-pharmacological nutraceutical. It is a common first-line nutraceutical used for patients with PTH insomnia; however, one study also investigated its potential role in treating PTH. In a prospective cohort trial of pediatric patients in a TBI clinic, 75% of patients reported significant improvements in headache symptoms after receiving administration of melatonin (3 mg and increased to 10 mg) [46]. The authors did not describe whether results could have been attributed to improvements in sleep. Several mechanisms linking melatonin to TBI and headache alleviation have been proposed, but more clinical evidence is needed [47].

Magnesium is a nutraceutical that has displayed both pain-reducing and neuroprotective effects in individuals recovering from TBI [32]. It has been suggested that magnesium may antagonize the effects of glutamate, an excitatory neurotransmitter that can be released after the occurrence of TBI [6]. Glutamate has been found to be elevated in patients with migraines, indicating a potential role of magnesium in the treatment of PTH [38]. Standiford et al. [48] conducted a two-arm randomized cohort study investigating the effects of oral magnesium in adolescent patients experiencing PTH following a concussion. Compared with the placebo group, which received 500 mg acetaminophen twice daily, the intervention group received an additional 400 mg of oral magnesium oxide twice a day. At the end of the study, the intervention group showed significant improvements in post-concussive symptom severity scores (PTH included) from baseline to 48 h after the administration of magnesium.

Another substance that has been suggested for use in patients with PTH are omega-3 fatty acids (O3FA). O3FAs have previously been investigated in the management of chronic daily headaches and migraines and have displayed clinically significant improvements in outcomes [49,50]. Currently, there are no published studies examining the use of O3FA in headache following TBI; however, there is an ongoing RCT on its use in veterans with PTH [51]. The authors hypothesize that dietary n−3 eicosapentaenoic acid (EPA) and docosahexaenoic acid (DHA) and n−3 linoleic acid (LA) can reduce PTH symptoms by interacting with lipid nociceptor mediators.

## 7. Neuromodulation

Repetitive transcranial magnetic stimulation (rTMS) is one of the most popular potential neuromodulation methods currently being investigated to treat PTH. It is a non-invasive technique that involves exposing a targeted region of the brain to repetitive magnetic impulses generated from an electrical current. Repetitive transcranial magnetic stimulation is FDA approved for the treatment of several central nervous system pathologies including treatment-resistant depression, migraine with aura and obsessive-compulsive disorder [52,53].

Several RCTs have demonstrated beneficial effects of rTMS in patients with PTH. Leung et al. [54] conducted a sham-controlled randomized study examining the intervention in 24 military veterans with PTH. The participants were randomized to either receive a total of 2000 pulses (10 Hz) of rTMS at the left motor complex (LMC) or a sham procedure. The intervention group demonstrated a statistically significant increase in the amount reduction of headache intensity when compared to the sham group at both 1-week and 4-week post-treatment. At the end of the study, there was an overall significant reduction in the headache exacerbation composite score in the rTMS group compared to the sham group. Another RCT conducted by Leung et al. [55] studied the use of rTMS in the left dorsolateral prefrontal cortex region in patients with mild TBI-related headache and depression. The group of participants receiving rTMS showed a statistically significant reduction in average headache intensity and headache prevalence compared to the control group at both 1-week and 4-week post-treatment.

Stilling et al. [56] conducted a more recent pilot, double-blinded RCT investigating the use of rTMS in adult patients experiencing PTH and post-concussion symptoms. In a study population of 24 participants, there was an observed time effect on headache severity and frequency at 1-month post-treatment; however, there was no significant difference in headache outcomes between the intervention and control groups.

Translingual neurostimulation (TLNS) is another neuromodulation technique that has been studied to treat PTH. This non-invasive method involves the electrical stimulation of the anterior surface of the tongue to activate the lingual branch of the trigeminal and the chorda tympani of the facial nerve. Translingual neurostimulation has been used in the rehabilitation of several central nervous system conditions including spinal cord injury and stroke [57].

Ptito et al. [58] conducted a multicenter, double-blind RCT examining the use of TLNS and physical therapy in the treatment of patients with PTH after mild-to-moderate TBI who had plateaued in recovery. Participants were randomized such that they received physical therapy and either a high-frequency or low-frequency pulse TLNS. Compared with the baseline, both groups demonstrated reductions in headache disability scores after 5-weeks post-treatment; however, results were not statistically significant. Another double-blind RCT comparing high- and low-frequency TLNS and physical therapy was conducted by Tyler et al. [57]. After 26 weeks of treatment, there was a 40% reduction in Headache Disability Index scores among both groups receiving the intervention, indicating lower headache frequency and severity.

Additional forms of neuromodulation have also been explored in the treatment of PTH. One study [59] investigated the use of infra-low neurofeedback (IF NFB), a modified type of biofeedback that focuses on extremely low-frequency brain waves (as low as 0.0001 Hz) that are responsible for core neuroregulatory networks. Through IF NFB, the participant’s brain receives a cue when dysregulatory brain frequencies are detected, prompting the brain to self-regulate and improve performance through associative learning [59]. The study investigated the use of this technique in veterans with diagnosed mTBI who were experiencing chronic headache. Participants received IF NFB 3 times per week for a total of 20 sessions. At the end of the study, there was a clinically significant reduction in the HIT-6 scores compared with the baseline.

While other potential cranial nerve targets exist, there is a lack of quality evidence showing their use in patients with PTH. One such example is vagal nerve stimulation (VNS), which is an FDA-approved technique used to treat cluster migraines and has been suggested as a potential treatment for TBI-related headaches. There are several proposed mechanisms linking VNS to PTH recovery (e.g., anti-inflammatory, cerebral edema reduction); however, more RCTs are needed [60].

Neuromodulation continues to be an intriguing treatment option for the management of PTH as shown in findings outlined in Table 2; however, more research is needed to determine how cranial nerve stimulation can affect headache outcomes in these patients.

## 8. Osteopathic Manipulation Treatment (OMT)

There is limited high-quality evidence demonstrating the use of osteopathic manipulative treatments (OMT) in the management of PTH [61,62]. In one pilot study [63], 10 military veterans with TBI and PTSD received 2 customized sessions of mixed manual therapies based on their pattern of tension and were evaluated for headache and anxiety outcomes. The intervention was performed by a certified massage therapist and primarily consisted of craniosacral therapy, Brain Curriculum, and neck petrissage. When compared to the baseline, participants demonstrated significant reductions in headache intensity scores after each of the two massage sessions. There was also a significant reduction in pain interference scores, an indicator of the negative impact pain has on someone’s life.

In another controlled pilot study [64], 26 patients experiencing headache secondary to post-concussion syndrome were assigned to either an OMT treatment group (13 patients) or a control group (13 patients) with no OMT. The specific OMT techniques chosen were muscle energy, myofascial release, counterstain and suboccipital release due to their popularity and lack of adverse effects. Patient data (visual analog score, HIT-6 scores) were collected immediately after the intervention and at 4-week and 6-week follow-up. At the end of the study, participants demonstrated statistically significant reductions in visual analog scale scores immediately after the intervention, indicating decreased pain intensity. While there was an observed improvement in the HIT-6 scores, the results were not statistically significant when comparing baseline to follow-up.

## 9. Interdisciplinary Treatments

Other approaches used to treat PTH may integrate several of the aforementioned techniques in an interdisciplinary approach. One study [65] incorporated a combination of neuropsychological treatment, exercise therapy, physiotherapeutic coaching and individualized social services. Eighty-nine participants were randomized into either an interventional or control group, and were assessed for post-concussion symptoms, headache outcomes, depression, fatigue and quality of life. Patients receiving the intervention took part in the interdisciplinary program for 22 weeks and demonstrated significant reductions in post-concussion symptoms and composite HIT-6 test scores. While more research involving these interventions are needed, interdisciplinary treatments may offer beneficial effects for patients with PTH.

## 10. Treatment Recommendations

Choosing a specific non-pharmacologic intervention for the management of PTH is a shared decision-making process between both the patient and the clinician. Non-pharmacologic intervention recommendations should be stratified based on relative ease of implementation. Given their high accessibility through dietary supplements and certain foods, nutraceuticals are recommended as part of the first-line non-pharmacologic treatment of PTH. While there is a lack of high-quality evidence supporting their use in PTH, nutraceuticals have demonstrated neuroprotective benefits in patients with migraine and have few adverse effects. Additionally, lifestyle modifications (e.g., sleep hygiene and physical activity) should always be incorporated into first-line treatment. They can be easily integrated into a patient’s daily routine and do not require physician oversight.

If first-line measures fail to improve PTH symptoms, cognitive behavioral therapy and acupuncture can be discussed with patients. CBT has demonstrated effectiveness in reducing PTH symptoms and headache-related disability. It is also time-intensive, requiring patients to attend regularly scheduled sessions, and may be best reserved for individuals experiencing psychosocial disturbances secondary to PTH. Regarding acupuncture, there is limited yet promising evidence supporting its use in the treatment of PTH. However, acupuncture’s relatively invasive nature may result in physical discomfort and can potentially exacerbate headaches and muscle pain [31]. Clinician–patient discussions are therefore necessary to assess the feasibility of these treatment options. If they can be afforded and are acceptable to the patient, CBT and acupuncture should be pursued as alternatives for patients actively seeking complementary therapies.

More intensive therapies may be considered in patients who continue to display debilitating symptoms despite attempts with multiple interventions. Several RCTs included in this review indicate the potential benefits of neuromodulation in the treatment of PTH. Despite these promising results, its use has several potential drawbacks. Neuromodulation requires that patients attend treatment sessions over the course of several weeks and months. The use of this intervention also carries several potential adverse effects including dizziness, seizures or changes in facial sensation [56]. Considering the benefits and risks of this intervention, neuromodulation may be an effective option in patients with refractory PTH.

## 11. Limitations and Future Directives

Given the number of TBI cases in the United States alone, there is a glaring need for formal evidence-based guidelines for the treatment of PTH. The absence of such guidelines has made the management of patients with PTH a challenge, raising uncertainty in which interventions to pursue. As more trials are conducted in the future, systematic reviews evaluating the results of these studies will be vital. Such research would aid in the creation of formal guidelines and provide clarity to clinicians in their efforts to treat patients with PTH.

A variety of non-pharmacological treatments have been described in this review, but current knowledge is mostly based on evidence from smaller-scale studies. More randomized controlled trials with larger sample sizes are needed to firmly establish evidence on the effectiveness of these interventions. Such trials would aid in the formation of an evidence-based algorithm for non-pharmacological treatments for PTH. This would allow clinicians to tailor individual treatment regimens for patients seeking non-pharmacological alternatives while also providing high-quality care.

While non-pharmacological interventions may be effective in the treatment of PTH, it is currently unclear how they compare to pharmacologic therapies. Future systematic reviews comparing the effectiveness of pharmacological and non-pharmacological treatments would be helpful to guide clinicians on incorporating both into the management of PTH. The hope is that this article will guide future research as there exists a need to establish formal evidence-based guidelines for the management of patients experiencing PTH.

## 12. Conclusions

Post-traumatic headache is one of the most common yet debilitating complications of TBI. The headache that develops in the time following injury can have major implications on an individual’s function and quality of life. Given the phenotypic variability and underdiagnosis of PTH, a high clinical suspicion and comprehensive history are vital in its diagnosis. Non-pharmacological interventions may offer an intriguing and effective alternative to analgesics such as NSAIDs in that they avoid any harmful side effects while also addressing a patient’s comorbidities. There are several options of treatment that can be chosen and tailored based on clinical judgment and a patient’s preferences and biopsychosocial needs. Although these non-pharmacological interventions are promising, this review emphasizes the need for more high-quality studies in order to provide clinicians with evidence-based guidance in the management of PTH.

## Figures and Tables

**Figure 1 brainsci-12-01357-f001:**
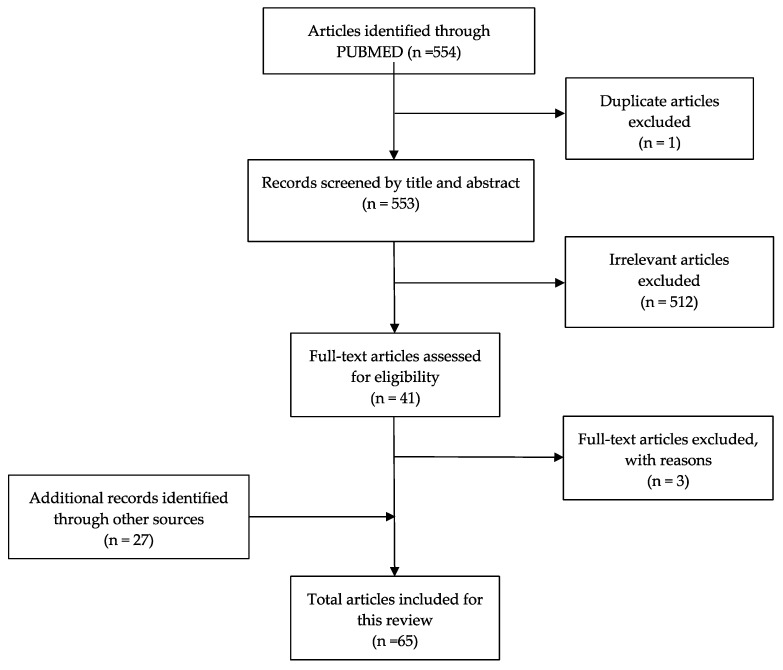
Process flow diagram.

**Table 1 brainsci-12-01357-t001:** Summary of types of non-pharmacological interventions used in management of PTH.

Intervention Type	Examples	Summary of Findings
Cognitive and Behavioral Modification	Cognitive Symptom Management and Rehabilitation Therapy (CogSMART) Progressive Muscle Relaxation (PMR) Cognitive Behavioral Therapy (CBT)	CogSMART demonstrated significant symptom improvement in patients with PTH PMR showed mixed results in PTH treatment but may be effective in reducing PTH incidence Several RCTs suggest potential benefit of CBT on PTH symptoms and headache disability
Acupuncture	Traditional Chinese Acupuncture (TCA) Auricular Acupuncture (AA)	Limited evidence but promising intervention One RCT indicates acupuncture may have significant benefits in PTH outcomes AA was demonstrated to be more effective than TCA in reducing PTH symptoms
Lifestyle Modification	Rehydration Nutrition Sleep hygiene	One study showed rehydration, nutrition and sleep hygiene modification reduced headache frequency, severity and improved quality of life More research examining individual modifications is needed
Physical Activity	Early exercise Cervicovestibular physical therapy	One study showed presence of PTH was significantly less in those who did early exercise compared to those did not Physical therapy showed promising results in reducing PTH symptoms
Nutraceuticals	Vitamin D Vitamin B2 (riboflavin) Coenzyme Q Melatonin Magnesium Omega-3 fatty acids (O3FA)	No recent trials examining vitamin D, riboflavin or coenzyme Q exist, but preclinical evidence suggests potential benefit Melatonin and magnesium showed significant reduction in PTH-related symptoms Ongoing trial investigating use of O3FAs, but preclinical mechanisms suggest benefit
Neuromodulation	Repetitive transcranial magnetic stimulation (rTMS) Translingual neurostimulation (TLNS) Neurofeedback Vagal nerve stimulation	Several RCTs demonstrated significant impact of rTMS in the improvement of PTH outcomes Use of TLNS showed mixed results, but larger RCTs are needed Neurofeedback used significantly reduced headache disability scores in veterans with PTH No studies examining vagal nerve stimulation, but may be potential intervention
Interdisciplinary	Integrative therapies (combination of psychosocial, physical and social interventions)	Combination of interventions may treat holistic needs of patients with PTH One study integrated neuropsychological, exercise, physiotherapeutic and social interventions; significant improvements in PTH symptoms and headache disability were shown

**Table 2 brainsci-12-01357-t002:** Summary of study findings of neuromodulation interventions for management of post-traumatic headache.

Study	Study Design	Intervention	Participants	Headache Outcome Measure(s)	Results	**Limitations**
Leung et al. (2016) [54]	Sham, randomized controlled trial	rTMS	rTMS Group: *n* = 12 Age: 18–80 years (Mean: 41.2 years) SHAM Group: *n* = 12 Age: 18–80 years (Mean: 41.4 years)	Headache exacerbation composite score (sum of headache intensity rating/duration)	The rTMS group showed significantly higher % reduction in headache intensity compared to SHAM group at 1-week post-treatment The rTMS had significantly lower headache composite score compared to SHAM group at 4-weeks post-treatment	Small sample size Study duration (1 month) does not address potential relapse of PTH in weeks following completion of treatment Composite score may not be representative of headache duration
Leung et al. (2018) [55]	Sham, randomized controlled trial	rTMS	rTMS Group: *n* = 14 Age: 18–65 years (Mean: 33 years) SHAM Group: *n* = 15 Age: 18–65 years (Mean: 35 years)	Average daily headache intensity Occurrences of daily headaches Debilitating headache composite score	Average daily intensity was significantly lower in the rTMS group compared to the SHAM group at 1-week and 4-weeks post-treatment rTMS group had significantly lower debilitating headache composite scores and frequency compared to SHAM group at 1-week and 4-weeks post-treatment	Depression is a common comorbidity in chronic pain and mTBI patients that can affect perception of headache intensity Small sample size
Stilling et al. (2020) [56]	Pilot, double-blinded randomized controlled trial	rTMS	rTMS Group: *n* = 10 Age: 18–65 years (Mean: 40.3 years) SHAM Group: *n* = 10 Age: 18–65 years (Mean: 31.6 years)	Headache severity and frequency Headache Impact Test-6 (HIT-6)	rTMS group showed a significant overall time effect for headache severity and decrease in headache frequency 1-month post-treatment No significant difference in headache outcomes between rTMS and SHAM groups	Small sample size Three subjects did not receive MRI for neuronavigation, which could have affected treatment plan and effects Participants were on varying medication regimens
Tyler et al. (2019) [57]	Double-blinded randomized controlled trial	Translingual neurostimulation	High-Frequency Group: *n* = 22 Age: 18–65 years (Mean: 54.1 years) Low-Frequency Group: *n* = 21 Age: 18–65 years (Mean: 53.2 years)	Headache Disability Index	Forty percent reduction in Headache Disability Index Scores in both high-frequency and low-frequency groups 26-weeks post-treatment	Reduction in headache scores could be due to time considering length of study (26 weeks) Variability of TBI Sex differences Absence of patient data Potential Hawthorne effect
Ptito et al. (2021) [58]	Double-blinded randomized controlled trial	Translingual neurostimulation + physical therapy	High-Frequency Group: *n* = 59 Age: 18–65 years (Mean: 48.9 years) Low-Frequency Group: *n* = 63 Age: 18–65 years (Mean: 43.8 years)	Headache Disability Index	Both high-frequency and low-frequency groups demonstrated reductions in Headache Disability Index Scores	Imbalance between female and male participants Difficult to attribute causation to neurostimulation due to addition of PT
Nelson et al. (2015) [59]	Pilot study	Infra-low neurofeedback	*n* = 4 Age: 18–60 years (mean: 42 years) Military veterans	Headache Impact Test-6 (HIT-6)	Clinically significant improvements in HIT-6 scores, sleep, quality of life and attention	Small sample size Lack of control group Time constraint to complete the study

## Data Availability

Not applicable.

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
