# Peer review of "Update on Non-Pharmacological Interventions for Treatment of Post-Traumatic Headache"

_brainsci, 2022, doi:10.3390/brainsci12101357_

Round 1
Reviewer 1 Report
This review provides an update on non-pharmaceutical interventions and clinical guidance for management of patients with PTH.
The introduction section provides a comprehensive background for the review.
I think it s important to address the effects of repeated head impacts that result from failure to diagnose or poor follow up resulting in premature return to play in a sporting context. Some sports, for example, have a minimum period of removal from play of only 7 days. This is relevant for the interventions you address later in the review.
The methods are clearly explained and Figure 1 is very helpful for the reader.
Lines 52, 112, 135, 146, 148, 162, 168, 170, 171, 172, 197, 259, 307, 322: Change "compared to" to "compared with"
Line 117-118: This sentence is unclear. Please rephrase. It seems as though the word "symptoms" may be missing from between "individuals" and "caused."
Lines 273 and 276: Change "rTMS" to "Repetitive transcranial magnetic stimulation (rTMS)."
Line 284: Change "%" to "amount."
Line 300 and 374: As for lines 273 and 276, it is best not to start a sentence with an acronym.
Reviewer 2 Report
1. Abstract. It would be interesting to provide some leaving messages (future directions) or more detailed data after describing the review aims.
2. Could the authors provide an abstract figure of the manuscript? This would significantly impact the quality of the manuscript.
3. The authors described the possible non-pharmacological interventions for managing post-traumatic headache. Could the authors provide a section specifically comparing the different therapies? Provide an algorithm for clinical practice. A discussion should be provided.
4. An important section for every review is what the authors observed as flaws in the literature that should be addressed in future studies. Could the authors provide a chapter about future directions?
Round 2
Reviewer 2 Report
The reviewer did not find the abstract figure.
Author Response
Point 1: The reviewer did not find the abstract figure.
Response 1: We thank the reviewer for making us aware of this issue. We discussed this with the editing team and learned that they did not send the figure previously. As the figure is a graphical abstract, it is a separate file from the revised manuscript document. We will submit the revised manuscript and graphical abstract again, and the figure should be delivered to the reviewer.